# LncRNA Taurine Up-Regulated 1 Knockout Provides Neuroprotection in Ischemic Stroke Rats by Inhibiting Nuclear-Cytoplasmic Shuttling of HuR

**DOI:** 10.3390/biomedicines12112520

**Published:** 2024-11-04

**Authors:** Xiaocheng Shi, Sha Liu, Yichun Zou, Hengping Wu, Jinyang Ma, Junbin Lin, Xin Zhang

**Affiliations:** 1Department of Neurological Rehabilitation, Zhongnan Hospital of Wuhan University, Wuhan 430071, China; shixiaocheng2024@163.com; 2Department of Neurosurgery, Zhongnan Hospital of Wuhan University, Wuhan 430071, China; zyichun@hotmail.com (Y.Z.); 17371738876@163.com (H.W.); mjy3138@163.com (J.M.); 3Brain Research Center, Zhongnan Hospital of Wuhan University, Wuhan 430071, China; sha.liu@whu.edu.cn; 4Department of General Practice, Zhongnan Hospital of Wuhan University, Wuhan 430071, China

**Keywords:** TUG1, ischemic stroke, neuroprotection, HuR, nucleoplasmic shuttling, neuronal apoptosis

## Abstract

**Background**: Long non-coding RNA taurine-upregulated gene 1 (TUG1) is involved in various cellular processes, but its role in cerebral ischemia–reperfusion injury remains unclear. This study investigated TUG1’s role in regulating the nucleocytoplasmic shuttling of human antigen R (HuR), a key apoptosis regulator under ischemic conditions. **Methods**: CRISPR-Cas9 technology was used to generate TUG1 knockout Sprague Dawley rats to assess TUG1’s impact on ischemic injury. The infarct area and neuronal apoptosis were evaluated using TUNEL, hematoxylin and eosin (HE), and TTC staining, while behavioral functions were assessed. Immunofluorescence staining with confocal microscopy was employed to examine TUG1-mediated HuR translocation and expression changes in the apoptosis-related proteins COX-2 and Bax. **Results**: TUG1 knockout rats showed significantly reduced cerebral infarct areas, decreased neuronal apoptosis, and improved neurological functions compared to controls. Immunofluorescence staining revealed that HuR translocation from the nucleus to the cytoplasm was inhibited, leading to decreased COX-2 and Bax expression levels. **Conclusions**: TUG1 knockout reduces ischemic damage and neuronal apoptosis by inhibiting HuR nucleocytoplasmic shuttling, making TUG1 a potential therapeutic target for ischemic stroke.

## 1. Introduction

Cerebral ischemia–reperfusion injury represents a critical pathological condition associated with ischemic stroke [1] in which the restoration of blood supply to the brain following a period of ischemia leads to exacerbated tissue damage rather than recovery [2]. This paradoxical phenomenon results from a complex cascade of biochemical and molecular events [3], including oxidative stress, inflammation, and neuronal apoptosis [4]. Among the myriad molecular players, RNA molecules, particularly long non-coding RNAs (lncRNAs), have recently emerged as significant regulators of gene expression in various diseases, including cerebral ischemia [5,6,7].

lncRNAs, which are longer than 200 bp and non-coding for proteins, are abundantly and differentially expressed in the nervous system across various developmental stages and under diverse conditions [8]. Taurine-upregulated gene 1 (TUG1), implicated in diverse biological processes including cell proliferation, apoptosis, and differentiation, has a poorly understood role in cerebral ischemia–reperfusion injury [5,9,10]. In ischemic stroke, the rs2240183 C allele of taurine-upregulated gene 1 (TUG1) enhances the risk by enabling the binding of the transcription factor GATA-1, thereby elevating TUG1 expression [11]. Concurrently, TUG1 aggravates neuronal apoptosis and necroptosis by sequestering miR-204 [12] and miR-145 [13] in ischemic stroke, thereby modulating apoptosis and inflammatory responses to promote further neuronal damage. Advances in proteomics and epigenetics have revealed a significant interaction network between lncRNAs and RNA-binding proteins (RBPs), which is crucial for disease development [14,15,16,17].

HuR, a nucleocytoplasmic shuttling protein, regulates the stability and translation of mRNAs from genes involved in cell survival and apoptosis [18,19,20], such as COX-2 [21,22] and Bax [23,24], enhancing their expression by translocating from the nucleus to the cytoplasm under ischemic conditions. Studies have shown that in neurodegenerative diseases such as amyotrophic lateral sclerosis (ALS), HuR plays a role in the chronic activation of microglia through co-transcriptional and post-transcriptional regulation of gene expression, highlighting its potential as a therapeutic target for these diseases [25,26]. The ectopic expression of HuR in astrocytes has been shown to worsen outcomes after transient ischemic stroke by increasing cerebral vascular permeability and edema [27,28], which highlights its potential as a therapeutic target in cerebral ischemia. Additionally, it is hypothesized that HuR influences neuronal survival and the response to ischemia in reperfused hippocampal neurons by modulating the stability and translation of mRNAs containing adenine- and uridine-rich elements (AREs). Emerging evidence indicates that TUG1 interacts specifically with HuR [29,30], potentially modulating its subcellular localization and function, which is critical under stress conditions such as ischemia–reperfusion injury. This interaction may influence HuR’s ability to stabilize mRNA and regulate translation, though the precise mechanisms remain unclear. Our research focuses on this interaction within the context of cerebral ischemia and how it affects cellular resilience and recovery. In our previous studies, we have demonstrated that TUG1 can enhance the transfer of HuR from the nucleus to the cytoplasm in HT22 cells under oxygen–glucose deprivation conditions, increasing COX-2 mRNA expression and accelerating apoptosis [31]. CRISPR-Cas9 technology was employed in this study to knock out the TUG1 gene in Sprague Dawley rats. It was hypothesized that without TUG1, the pathological relocation of HuR is hindered, thus reducing neuronal damage and improving post-ischemic outcomes. Through a series of experiments, including gene knockout, histological assessments, and behavioral tests, we aimed to uncover the protective role of TUG1 silencing in ischemic stroke, potentially offering new therapeutic avenues targeting lncRNAs.

TUG1’s significant interaction with HuR has been studied through a combination of computational prediction and experimental validation [29,30]. Researchers first employed bioinformatic tools, such as RPISeq, RBPmap, and RBPDB, to predict potential HuR binding sites on TUG1, which provided a theoretical basis for subsequent experimental studies. RNA immunoprecipitation (RIP) assays were then performed using HuR-specific antibodies to co-precipitate HuR and its bound RNA from cell lysates. qRT-PCR analysis showed that TUG1 was significantly enriched in the HuR complex compared to that in the negative control, indicating a direct interaction between them. Furthermore, RNA pull-down assays using biotin-labeled TUG1 probes captured HuR proteins bound to TUG1 from cell extracts, and Western blot analysis further confirmed this interaction. These findings demonstrate a specific interaction between TUG1 and HuR, providing a crucial basis for understanding the role of TUG1 in regulating HuR and its downstream signaling pathways.

## 2. Materials and Methods

### 2.1. Animal

Male Sprague Dawley rats (250–280 g) were housed in pairs or trios under a 12 h light/dark cycle at 23–25 °C, with free access to food and water. All procedures were approved by the Animal Care and Ethics Committee of Wuhan University School of Medicine (Approval Code: AF268, Approval Date: 6 April 2020) and followed the guidelines of the Institutional Animal Care and Use Committee (IACUC), as well as the ARRIVE guidelines. Animals were randomly assigned to groups, and experiments were conducted by blinded investigators.

### 2.2. Middle Cerebral Artery Occlusion (MCAO)

Male SD rats (8–12 weeks old, n = 50, approximately 280 g) were anesthetized with 4% isoflurane (RWD, R510-22) delivered through a face mask using a gas mixture of 70% nitrogen and 30% oxygen. A midline neck incision was made to carefully expose the right external carotid artery (ECA). An occlusion suture (Jialing, Nanchong, China, L3600) was advanced from the ECA into the right internal carotid artery to occlude the origin of the right middle cerebral artery (MCA) for 120 min. After the occlusion period, the suture was withdrawn to allow reperfusion, the ECA was ligated, and the incision was closed. In the sham-operated group, identical procedures were performed; however, the suture was inserted and immediately removed without causing occlusion. Rectal temperature was maintained at 37.0 ± 0.5 °C using a heating pad.

### 2.3. HT22 Cell Culture and Oxygen–Glucose Deprivation (OGD)

HT22 cells from a mouse hippocampal neuronal line were cultured in DMEM (Gibco, Grand Island, NY, USA, 11965092) supplemented with 10% FBS (Gibco, Grand Island, NY, USA, 10099141C) and 1% penicillin–streptomycin (Gibco, Grand Island, NY, USA, 15140122) in a 5% CO_2_ atmosphere. In oxygen–glucose deprivation (OGD) experiments, the cells were rinsed three times with EBSS (Gibco, Grand Island, NY, USA, 24010043) lacking serum and glucose, followed by incubation in EBSS at 37 °C in a 95% N_2_ and 5% CO_2_ atmosphere for 6 h. Concurrently, control cells were maintained under normal conditions for the same duration.

### 2.4. Modified Neurological Severity Score (mNSS)

The rats were assessed while awake using the mNSS to confirm the success of ischemic stroke induction, following established protocols. The mNSS test was administered again to all rats on the first day after MCAO.

### 2.5. Triphenyltetrazolium Chloride (TTC) Staining

Twenty-four hours post-reperfusion, rats were deeply anesthetized with isoflurane (RWD, R510-22). Cardiac perfusion was then performed with 150 mL of 0.9% saline, followed by rapid extraction of the brain. Each brain was sectioned into eight slices, incubated in 2% triphenyltetrazolium chloride (Sigma-Aldrich, St. Louis, MO, USA, T8877) at 37 °C for 30 min, and subsequently fixed in 4% paraformaldehyde (Servicebio, Wuhan, China, G1101) at 4 °C overnight.

### 2.6. CRISPR-Cas9

TUG1−/− rats were created with a Sprague Dawley background using CRISPR-Cas9 to delete a 9486 bp region, with two pairs of gRNAs (Table 1), validated by sequencing. Cas9 mRNA and gRNAs were then co-injected into fertilized eggs to produce the knockout rats. Specifically, two purified guide RNAs (50 ng/μL) and a Cas9 protein (40 ng/μL) were microinjected into the pronuclei of fertilized, one-cell, SD rat embryos. In this study, a total volume of 2.0 nL containing gRNA and Cas9 protein was microinjected into each embryo. The volume was chosen to maintain a weight-to-volume ratio of approximately 1:100 to 1:150, ensuring embryo viability. After injection, embryos were cultured for 24–48 h until reaching the blastocyst stage. A total of 100 embryos were used for microinjection, with 80 embryos surviving after the procedure. The surviving embryos were then surgically implanted into the oviducts of pseudo-pregnant SD females. Pseudo-pregnancy was induced by mating female rats with vasectomized males. Embryo transfer was performed following standard protocols. The pups were genotyped via PCR followed by sequence analysis to confirm successful editing. Offspring were genotyped and further bred to obtain homozygous TUG1 knockouts.

### 2.7. DNA and Quantitative Real-Time Reverse Transcription PCR (qPCR)

Genomic DNA was extracted using the TaKaRa MiniBEST Universal Genomic DNA Extraction kit (Ver. 5.0, Code No. 9765) to ensure high purity. PCR was performed in a total volume of 30 µL, containing all primers listed below, using LA Taq DNA polymerase (TaKaRa, Shiga, Japan, RR02AG) for amplification. The reaction was subjected to the following conditions: an initial denaturation at 94 °C for 3 min, followed by 35 cycles of 94 °C for 30 s (denaturation), annealing at 60 °C for 30 s (depending on the primer Tm), and extension at 72 °C for 1 min (adjusted based on product length). A final extension was performed at 72 °C for 5 min, and the reaction was then held at 4 °C. Two controls were included in the PCR genotyping: a water control with no DNA template added, and a wild-type control with 400 ng of rat genomic DNA. The primer sequences were as follows in Table 2, and were synthesized by Sangon Biotech. Table 3 shows the PCR amplification products for TUG1 genotyping.

### 2.8. Immunostaining

Brain tissues were fixed, embedded in paraffin, and sectioned at a thickness of 5 μm using a Leica SM2010R microtome. Sections underwent deparaffinization, followed by antigen retrieval in citrate–EDTA buffer. Afterward, the sections were washed in PBS and blocked with a solution containing 0.1% Triton X-100 and 10% serum for 1 h at room temperature. Primary antibody incubation was performed overnight at 4 °C with the following antibodies: COX-2 (ABclonal, Wuhan, China, A22871) at a 1:1000 dilution, BAX (Cell Signaling, Danvers, MA, USA, 14796S) at a 1:800 dilution, and HuR (ABclonal, Wuhan, China, A19622) at a 1:1200 dilution. After incubation, sections were washed three times in PBS (5 min per wash) to remove excess primary antibodies. Subsequently, the sections were incubated with Alexa Fluor 594-conjugated secondary antibodies (ABclonal, Wuhan, China, AS039) at a 1:1000 dilution for 1 h at room temperature in the dark. Following the secondary antibody incubation, sections were washed again in PBS three times (5 min per wash) to ensure thorough removal of unbound secondary antibodies. Finally, the sections were mounted with Prolong Diamond Anti-Fade mountant (ABclona, Wuhan, Chinal, AS014) and allowed to cure for at least 24 h before imaging.

### 2.9. Confocal Imaging of Cytoplasmic Transfer

Fluorescence imaging was conducted using a Leica Stellaris 5 WLL confocal microscope equipped with a white light laser (WLL) for flexible wavelength selection. DAPI-stained nuclei were excited at 405 nm, with emission detected between 420 and 480 nm. The target protein was labeled with Alexa Fluor 555 and excited at 561 nm, and the emission was measured between 570 and 620 nm. An acousto-optical beam splitter (AOBS) was utilized to minimize spectral overlap, ensuring specific signal detection. All images were captured using a 63× oil immersion objective (NA = 1.4) to achieve optimal resolution. Imaging parameters, including laser power (20–25%), gain, and exposure time, were initially optimized using control samples to minimize photobleaching and reduce background noise while maximizing signal intensity. Z-stack images were acquired at 0.5 μm intervals to comprehensively cover both nuclear and cytoplasmic regions. Each image was averaged over four scans to enhance the signal-to-noise ratio. Identical acquisition settings were applied to all samples to enable reliable comparative analysis.

Following image acquisition, the ImageJ software was used for analysis. Images were first processed by subtracting the background using the “Subtract Background” function with a rolling ball radius of 50 pixels to reduce non-specific signals. The “Color Threshold” function was then used to set thresholds based on DAPI signals, allowing segmentation of nuclear and cytoplasmic regions. A binary mask was created for each channel, and the “Watershed” function was used to separate touching cells if needed. Subsequently, the “Analyze Particles” tool was employed to quantify the fluorescence intensity of the target protein within both nuclear and cytoplasmic compartments, with a size filter set to 100–10,000 pixels to exclude artifacts. For each cell, the average fluorescence intensity was measured in both nuclear and cytoplasmic regions, and the ratio of nuclear-to-cytoplasmic fluorescence was calculated. The total fluorescence intensity across the entire cell was also determined. All measurements were repeated across multiple fields of view (*n* = 5) for each condition to ensure sufficient statistical power for downstream analysis.

### 2.10. Hematoxylin and Eosin (HE)

Rat brains were sectioned coronally at a thickness of 10 µm and mounted onto glass slides. Tissue sections were deparaffinized in xylene (Servicebio, Wuhan, China, G1003) and rehydrated through a graded ethanol series to water. Sections were stained with hematoxylin (Servicebio, Wuhan, China, G1004) for 5 min, followed by rinsing in running tap water to remove excess stain. Subsequently, sections were counterstained with eosin (Servicebio, Wuhan, China, G1001) for 2 min and rinsed again with running tap water. After staining, sections were dehydrated through graded ethanol, cleared in xylene, and coverslipped with mounting medium (Servicebio, Wuhan, China, G1401).

Regions of interest (ROIs), such as the hippocampus and cortical areas, were identified under 10× magnification, and neuronal cells were manually counted at 40× magnification based on cell size, morphology, and nuclear staining characteristics. To reduce bias, three non-overlapping fields were selected per ROI, and the average neuronal count per ROI was calculated.

### 2.11. TdT-Mediated Biotin-16-dUTP Nick-End Labeling (TUNEL) Assay

Apoptosis was assessed using a TUNEL assay kit (Beyotime, Shanghai, China, C1090) following the manufacturer’s protocol. Tissue sections (4–5 µm thick) were deparaffinized in xylene, rehydrated through a graded ethanol series, and incubated with proteinase K (20 µg/mL, Beyotime, Shanghai, China, ST533) for 20 min at room temperature. After fixation with 4% paraformaldehyde, sections were incubated with TdT enzyme and biotin-16-dUTP at 37 °C for 60 min. Following this, sections were treated with streptavidin–HRP, and apoptotic cells were visualized using DAB staining (Beyotime, Shanghai, China, P0203), resulting in brown staining. Hematoxylin (Beyotime, Shanghai, China, C0105) was used for counterstaining before dehydration, clearing, and coverslipping with neutral balsam (Beyotime, Shanghai, China, C0167). TUNEL-positive cells were visualized under a light microscope, and ImageJ software v1.53e was used for digital quantification. The apoptotic index (AI) was calculated as the percentage of TUNEL-positive cells relative to the total number of cells in each field.

### 2.12. Fluorescence In Situ Hybridization (FISH)

Tissue sections were fixed in 4% paraformaldehyde for 10 min at room temperature, followed by permeabilization with pre-cooled permeabilization solution for 5 min at 4 °C. After washing with PBS, sections were pre-hybridized with buffer at 37 °C for 30 min to reduce non-specific binding. Hybridization was performed overnight at 37 °C using a pre-heated, probe-containing buffer in a humidified chamber with the TUG1 fluorescent probe sequence ACAUCAUGAUGGCUGAAUGCUUCUU. The RiboTM lncRNA FISH Probe Mix, developed by RiboBio, incorporates innovative improvements in lncRNA-specific probe design and labeling, significantly enhancing probe sensitivity. Using the RiboTM Fluorescent In Situ Hybridization Kit, this method enables precise localization of low-abundance lncRNA transcripts within cells. Post-hybridization, sections underwent sequential washes with Hybridization Wash Solutions I, II, and III at 42 °C to remove unbound probes, followed by a final PBS wash. Signal detection involved an anti-fluorescein antibody linked to Alexa Fluor, enhancing TUG1 visualization under a fluorescence microscope. Scrambled probes were used as controls to confirm signal specificity, and lncRNA distribution was clearly observed using laser confocal microscopy.

### 2.13. Statistical Analysis

Data analysis was performed using GraphPad Prism 8, employing either an unpaired two-tailed t-test or one-way ANOVA. Graphs were generated in GraphPad Prism 8, with bars representing the range of values. The results are presented as mean ± standard deviation, with statistical significance set to a *p*-value of less than 0.05.

## 3. Results

### 3.1. Generating and Validating TUG1 Knockout Rats Using CRISPR/Cas9

TUG1 has been previously studied both in vivo and in vitro to confirm its role in promoting apoptosis in cerebral ischemia. To further validate its function, we generated TUG1 knockout (TUG1^KO^) rats. The rat TUG1 gene (Gene ID: 100125371; GenBank accession number: NR_130147.1) is located on chromosome 14 and comprises three exons. To silence TUG1, we designed a pair of guide RNAs (gRNAs) targeting exons 1 to 3 for use in the CRISPR/Cas9 system (Figure 1A). Cas9 mRNA and gRNAs were co-injected into fertilized eggs to induce gene editing, generating homozygous TUG1 knockout rats (TUG1 −/−). These knockout rats were then mated with wild-type (WT) rats (TUG1+/+) to produce TUG1+/− heterozygous offspring, which were exclusively derived from the progeny of homozygous and WT rats. Experimental verification was conducted exclusively on TUG1 −/− rats. Three sets of primers were designed (Figure 1B) to target the DNA for PCR amplification, and gel electrophoresis (Figure 1C–F) was performed to confirm successful TUG1 knockout. Finally, RNA was extracted from rat brain tissues to quantify TUG1 expression, thereby validating the knockout efficiency of TUG1 knockout (Figure 1G).

### 3.2. TUG1 Knockout Could Inhibit Apoptosis in Rats

To compare the behavioral and histological differences in brain tissue ischemia between TUG1^KO^ rats (TUG1+/−) and WT rats, we performed experiments on middle cerebral artery obstruction in the rats. We found that compared to wild-type rats, TUG1+/− heterozygous rats showed a significant reduction in neurobehavioral scores (Figure 2A), along with a significant reduction in brain tissue infarct area (Figure 2B,C) after 24 h of MCAO, indicating that TUG1 deficiency ameliorates the area of ischemic injury in brain tissue as well as neurobehavioral deficits in rats. In addition, TUNEL staining (Figure 2D,E) and HE staining (Figure 2F,G) of rat brain tissue sections showed that the proportion of apoptotic neurons and neuronal loss was smaller in TUG1+/− heterozygous rats than in WT rats after MCAO, indicating that TUG1 attenuated neuronal ischemic injury and apoptosis (Figure 2F,G).

### 3.3. TUG1 Mediates Subcellular Translocation of HuR

Our previous results showed that TUG1 mediates the translocation of HuR from the nucleus to the cytoplasm of HT22 cells under OGD stimulation conditions [31]. To investigate whether a similar mechanism also occurs in the MCAO rat model, we examined the distribution ratio of HuR in different subcellular compartments of the hippocampal CA1 neurons in TUG1 knockout rats (TUG1+/−) after MCAO using an immunofluorescence assay.

The immunofluorescence results revealed significantly increased HuR enrichment in the cytoplasm of CA1 cells in the MCAO group of wild-type rats compared to the sham group. On the contrary, no significant changes were observed in HuR enrichment in the cytoplasm of CA1 cells within the MCAO group of TUG1 knockout rats compared to the sham group (** *p* < 0.01, Figure 3A,B). These data were further supported by observations made using confocal microscopy (Figure 3C). Further analysis showed that the level of TUG1 enrichment in the nucleus of HT22 cells decreased by approximately 50% under hypoxia conditions compared to normoxia (>0.76-fold reduction). This suggests that TUG1 plays a role in promoting the cytoplasmic translocation of HuR under hypoxic conditions (Figure 3D,F). Based on these findings, we confirmed that TUG1 can facilitate the cytoplasmic translocation of HuR under ischemic conditions.

### 3.4. TUG1 Was the Key lncRNA for HuR Cytoplasmic Translocation and Regulation of Apoptotic Protein Expression in Hypoxia

Previously, we demonstrated the binding of HuR to COX-2 mRNA, which stabilizes and increases the expression of COX-2 in HT22 cells. Here, quantitative analysis of COX-2 expression in hippocampal neurons was performed using immunofluorescent staining in the MCAO rat model. We discovered that in TUG1 knockout rats (TUG1+/−) with less HuR translocated to the cytoplasm under MCAO conditions for 24 h, mediated by TUG1, the expression of the pro-inflammatory protein COX-2 was significantly reduced compared to in WT rats (Figure 4A). On the other hand, the expression of another pro-apoptotic regulator, Bax, which has been reported to have binding sites for HuR, showed a consistent trend with COX-2. This demonstrates that HuR can enhance COX-2 and Bax mRNA stability and their expression.

## 4. Discussion

In our study, the interaction between lncRNA TUG1 and the RNA-binding protein HuR was identified as a critical response mechanism to hypoxic stress, promoting the translocation of HuR from the nucleus to the cytoplasm. This relocation facilitates cellular adaptation to hypoxia by modulating the stability and translation of specific mRNAs, which are pivotal in determining cell fate under stress conditions [32,33]. Notably, HuR enhances the stability of COX-2 and Bax mRNAs, which are key players in the apoptotic pathways [27,34], thus amplifying their expression through the TUG1/HuR axis. The upregulation of these proteins activates apoptotic signaling cascades, as evidenced by changes in mitochondrial membrane permeability and the subsequent release of cytochrome c, culminating in apoptosis [35]. The elucidation of the roles of TUG1 and HuR in apoptosis not only enhances our understanding of the molecular underpinnings of ischemic stroke but also lays the groundwork for developing therapeutic strategies targeting HuR, potentially improving outcomes and quality of life for stroke patients.

The functional role of lncRNA TUG1 in ischemic stroke has been extensively explored through different mechanisms across various studies highlighting its involvement in neuronal damage and apoptotic pathways. Previous research has primarily focused on TUG1’s interaction with miRNAs, such as miR-204-5p [12] and miR-145 [13], through which it functions as a miRNA sponge to regulate the expression of genes involved in inflammation and apoptosis, adversely affecting recovery after stroke. These studies confirm the critical role of TUG1 in ischemic stroke, particularly in regulating neuronal injury and death, and our study supports this conclusion by highlighting the deleterious role of TUG1 in post-stroke recovery. In contrast to previous studies focusing on microRNA sponge functions, our results suggest that TUG1 acts by binding to HuR, thereby affecting the movement of HuR between nuclear and cytoplasmic compartments. This interaction greatly affects the stability and translation of key mRNAs involved in apoptosis and survival, such as COX-2 and Bax. This new understanding of the TUG1 mechanism not only advances our understanding of stroke pathophysiology but also offers new therapeutic possibilities to improve patient prognosis.

The present study examined the role of HuR, a protein known for regulating gene expression, in cellular functions beyond its recognized involvement under stress conditions such as hypoxia [18,21,36]. HuR is vital in post-transcriptional regulation as it binds to specific RNA sequences in the 3’UTR of mRNAs [37], thereby enhancing the stability and translation of these messengers. Key proteins involved in apoptosis, such as COX-2 and Bax, are influenced by HuR, particularly through its shift in location from the nucleus to the cytoplasm in response to stress [38,39], which impacts mRNA stability and protein production. HuR’s role in stabilizing the mRNAs of apoptosis-related proteins is especially significant in the context of neuronal damage [27,36,40]. This stabilization boosts the levels of COX-2 and Bax under stress conditions such as hypoxia, facilitating apoptosis. COX-2’s function in apoptosis is dual; it can promote or inhibit the process based on the cellular environment and external stimuli [41,42]. This research positions HuR’s regulatory dynamics as a potential target for therapy in diseases marked by abnormal apoptosis, such as neurodegenerative disorders. Modulating HuR’s activity or its interactions with specific mRNAs might offer new therapeutic approaches to managing neuronal apoptosis, providing potential treatments for conditions such as ischemic stroke.

Currently, effective treatments to prevent neuronal loss in stroke cases are lacking [43]. Our data suggest that inhibiting TUG1 or HuR can reduce neuronal apoptosis and slow down COX-2 or Bax-mediated apoptosis, highlighting the potential of targeting the TUG1/HuR/Bax pathway. Future clinical directions should explore TUG1 inhibition through antisense oligonucleotides (ASOs) or small interfering RNA (siRNA) to reduce neuronal damage by disrupting its interaction with HuR. Additionally, small molecule inhibitors or antibodies targeting HuR should be investigated with the aim of decreasing neuroinflammation and cell death by interfering with HuR’s mRNA-stabilizing function. Developing these therapeutic agents requires rigorous preclinical testing to ensure efficacy and safety, including optimizing delivery methods to effectively cross the blood–brain barrier.

However, the molecular mechanisms by which TUG1 influences HuR subcellular localization remain incompletely understood. One of the challenges we faced was the unsuccessful synthesis of wild-type/mutant TUG1-ARE lncRNA probes. This difficulty arose because AREs (AU-rich elements) are common not only in TUG1, but also in COX-2 mRNA and Bax mRNA. Mutating over 50% of the AREs might lead to structural changes in TUG1, rendering the probes ineffective. Conversely, our experimental results indicate no significant differences compared to the wild type if only a few AREs are mutated. We hypothesize that TUG1 functions as a scaffold in the cytoplasmic distribution process of HuR. Alternatively, TUG1 might control the subcellular localization of HuR through post-translational modifications of key structural domains such as RRM1, RRM2, and HNS. These hypotheses open new avenues for understanding the intricate regulation of HuR by TUG1, suggesting that TUG1 could modulate HuR’s interaction with its target mRNAs and influence cellular stress responses. Future research should employ advanced techniques such as RNA pull-down assays and cross-linking immunoprecipitation (CLIP) to map the precise interaction sites between TUG1 and HuR. Additionally, exploring the effects of specific inhibitors of post-translational modifications on HuR’s RNA-binding domains will be crucial. These studies will enhance our understanding of the TUG1-HuR regulatory axis and its potential as a therapeutic target in ischemic stroke. By addressing these mechanisms, we can develop more precise strategies to modulate HuR activity and improve therapeutic outcomes for stroke patients.

In this study, we opted to use TUG1 heterozygotes (TUG1+/-) instead of homozygous mutants (TUG1−/−) primarily to avoid severe physiological phenotypes or embryonic lethality that might result from a complete knockout. During breeding, we observed that a full knockout of TUG1 disrupted its critical functions in reproductive, developmental, or neurological systems, leading to noticeable neurological impairments, particularly in older animals, such as drowsiness and stereotypic behavior. Additionally, TUG1 knockout resulted in infertility in both male and female rats, consistent with previous studies showing reduced sperm quality in male rats lacking TUG1 [44]. In contrast, the use of the heterozygous model allowed for sufficient functional preservation of TUG1 under conditions of reduced expression, enabling normal growth and development while minimizing the interference of non-specific effects. A complete knockout of TUG1 might trigger compensatory mechanisms or lead to significant alterations in other molecular pathways, potentially masking or confounding the direct interaction between TUG1 and HuR. The heterozygous model maintains a certain level of TUG1 function, thereby reducing these non-specific effects and providing clearer and more accurate mechanistic data, allowing us to focus more precisely on the role of TUG1 under specific stress conditions.

Our study reveals a neuroprotective role of the long non-coding RNA TUG1 in ischemic stroke, specifically by reducing neuronal apoptosis through the inhibition of HuR nucleocytoplasmic translocation. This finding is an interesting contrast to previous research on TUG1 function. In certain pathological conditions, such as vascular cognitive impairment (VCI), high expression of TUG1 has been reported to correlate with increased neuronal apoptosis, and the knockdown of TUG1 alleviated neuronal damage and improved cognitive function [45]. This functional discrepancy may stem from the multifaceted roles of TUG1 in different tissues and pathological states.

On the one hand, TUG1 may mediate various signaling pathways by interacting with different molecular partners. In our study, TUG1 exerted neuroprotective effects by influencing HuR nucleocytoplasmic translocation, thereby regulating the expression of the downstream apoptosis-related proteins COX-2 and Bax. On the other hand, in studies concerning VCI, TUG1 is thought to promote neuronal apoptosis through interacting with brain-derived neurotrophic factor (BDNF). This suggests that TUG1 may possess dual functions in acute and chronic ischemia–hypoxia conditions depending on its interacting molecules and the type of pathology.

Future research should aim to further elucidate the molecular mechanisms of TUG1 in different disease contexts, particularly its interactions with key molecules such as HuR and BDNF. Additionally, considering the multifaceted roles of TUG1 in neurological diseases, developing personalized therapeutic strategies targeting TUG1 may hold significant clinical significance.

## 5. Conclusions

TUG1 knockout reduces ischemic damage and neuronal apoptosis by inhibiting HuR nucleocytoplasmic shuttling, making TUG1 a potential therapeutic target for ischemic stroke. These findings provide a basis for future therapeutic strategies targeting the TUG1–HuR axis to improve stroke recovery.

## Figures and Tables

**Figure 1 biomedicines-12-02520-f001:**
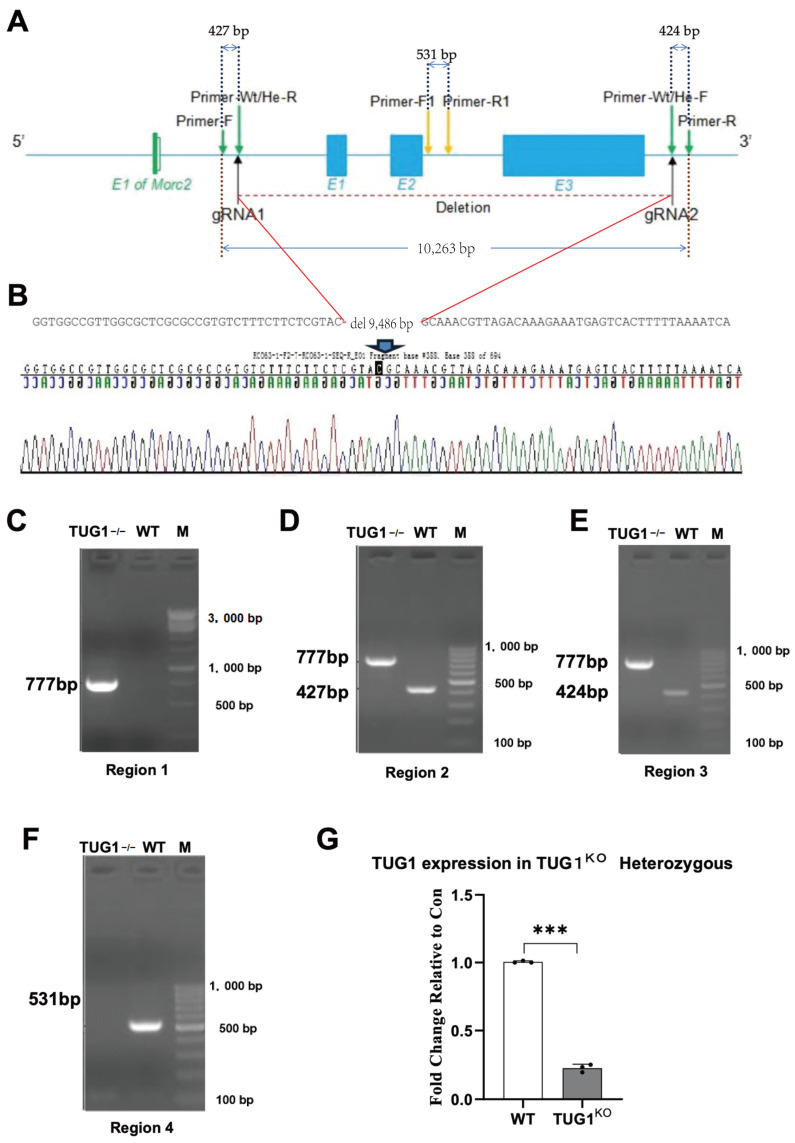
Generation and validation of the TUG1 knockout (TUG1^KO^) rat model. (**A**) Schematic representation of the rat TUG1 gene locus, including PCR primer locations and gRNA target sites (the gene orientation is from left to right; the total size is 7.04 kb). Solid boxes indicate open reading frames (ORFs), and open boxes represent untranslated regions (UTRs). (**B**) Sequencing verification of the TUG1^KO^ rats (−/−). Using gRNA1 and gRNA2, Cas9-mediated cleavage generated a 9468 bp deletion, encompassing all exons of the TUG1 gene. (**C**–**F**) Genotyping of TUG1^KO^ rats (−/−) by DNA gel electrophoresis of PCR-amplified products. Lane M: DNA size marker; TUG1−/−: homozygous knockout (−/−); lane WT: wild type (+/+). Primers used: (**C**) Tug1-F and Tug1-R; (**D**) Tug1-F and Tug1-R with Tug1-He/Wt-R; (**E**) Tug1-F with Tug1-He/Wt-F and Tug1-R; and (**F**) Tug1-F1 and Tug1-R1. (**G**) Quantitative real-time PCR analysis of TUG1 expression in TUG1^KO^ rats (TUG1+/−) showed a 73.7% reduction compared to wild-type (WT) controls. (*** *p* < 0.001).

**Figure 2 biomedicines-12-02520-f002:**
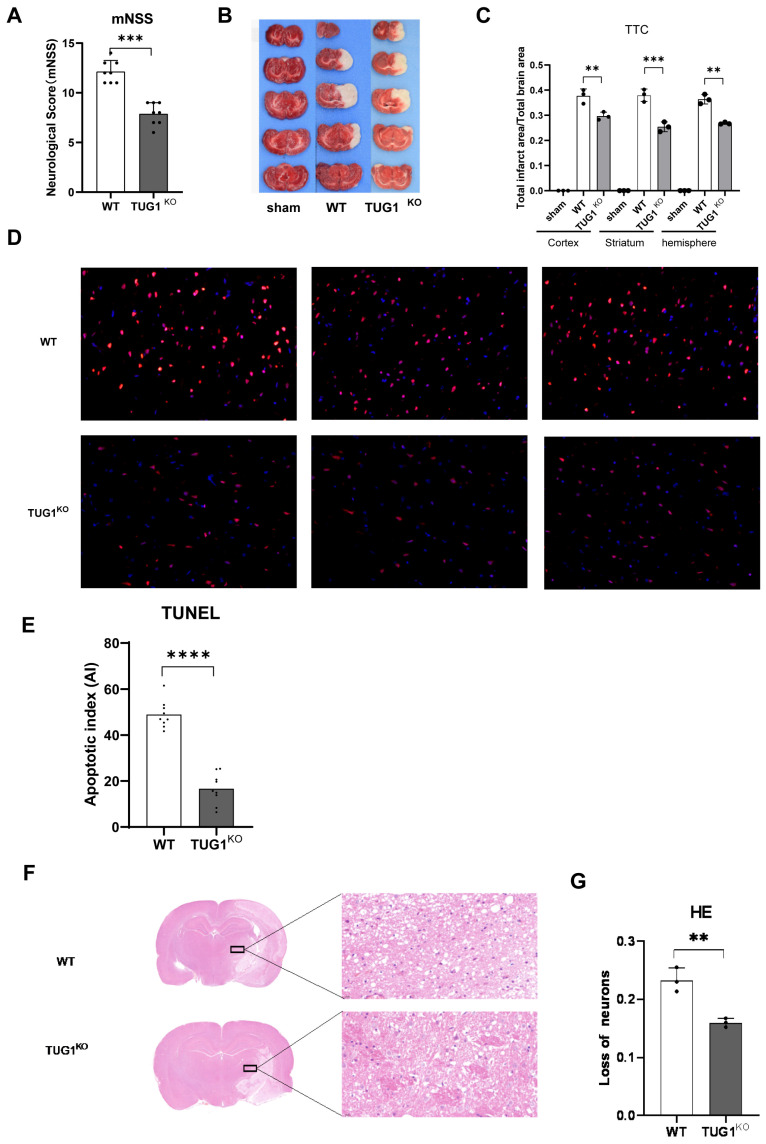
Knockout TUG1 reduced cerebral infarction and improved neurological outcome after MCAO. (**A**) Comparative analysis of mNSS scores for neurological deficits in rats 24 h after MCAO surgery. *** *p* < 0.001 compared to WT + MCAO, n = 8/group. (**B**,**C**) TTC staining of brain tissue sections after MCAO. The area of the infarct zone on representative images of coronal sections of the brain shown in white-stained areas and quantification of brain infarct volume is provided. ** *p* < 0.01 compared to WT + MCAO, n = 3/group. (**D**,**E**) Neuronal apoptosis revealed with TUNEL staining (×400) in the ischemic brain of rats and quantification of the number of positive apoptotic cells (red). The number of positive TUNEL cells was significantly lower in TUG1+/− heterozygous rats than in WT rats.(**** *p* < 0.0001) (**F**,**G**) HE staining (×100) was used to show changes in neurons, and more normal neurons were observed in TUG1+/− heterozygous rats than in WT rats, indicating a milder ischemic brain injury.

**Figure 3 biomedicines-12-02520-f003:**
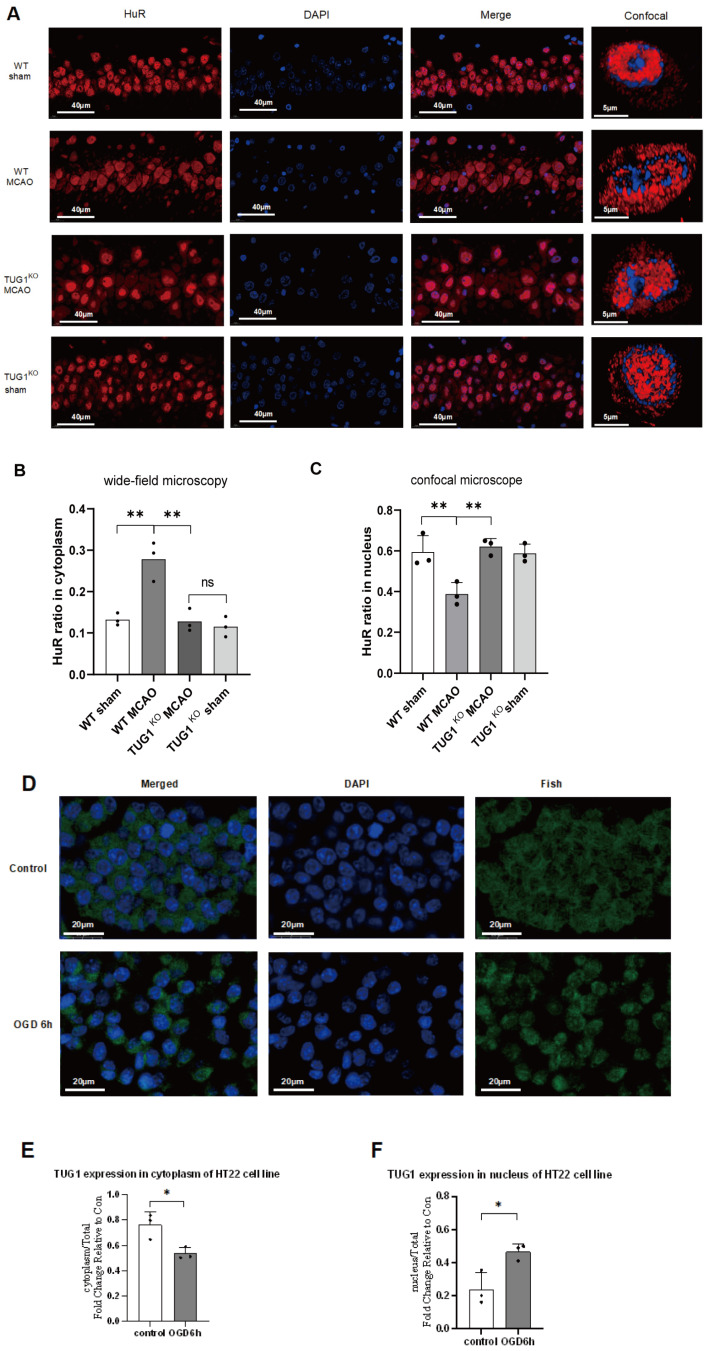
TUG1 promotes HuR cytoplasmic translocation. (**A**) Hippocampal CA1 neurons were imaged using wide-field microscopy and confocal microscopy. The red fluorescent region, indicated by arrows in the image, represents cytoplasmic HuR staining. The scale bar corresponds to 20 µm. (**B**) Immunofluorescence staining revealed that in WT rats after MCAO, the proportion of cytoplasmic HuR increased significantly compared to the overall cell proportion. In contrast, TUG1+/− heterozygous rats showed no significant difference in this proportion in comparison to the control group (** *p* < 0.01). (**C**) Confocal microscopy provided consistent results with wide-field microscopy (*n* = 5) (** *p* < 0.01). (**D**) Following 6 h of OGD treatment, HT22 cells were subjected to TUG1-FISH staining on paraffin-embedded sections. In these images, TUG1 appears in green, while the cell nuclei are stained with DAPI in blue. The green fluorescent region indicated by arrows in the control group denotes TUG1 staining in the cell nucleus. Conversely, the green fluorescent region indicated by arrows in the OGD group depicts TUG1 staining in the cytoplasm outside the cell nucleus. The scale bar represents 20 µm. (**E**,**F**) In HT22 cells, after 6 h of OGD treatment, the proportion of TUG1 in the cytoplasm decreased significantly compared to the control group from 76.19% to 53.43% (* *p* < 0.05), while the proportion of TUG1 in the cell nucleus increased significantly from 23.82% to 46.57% (* *p* < 0.05). Values for all panels are means ± standard deviations.

**Figure 4 biomedicines-12-02520-f004:**
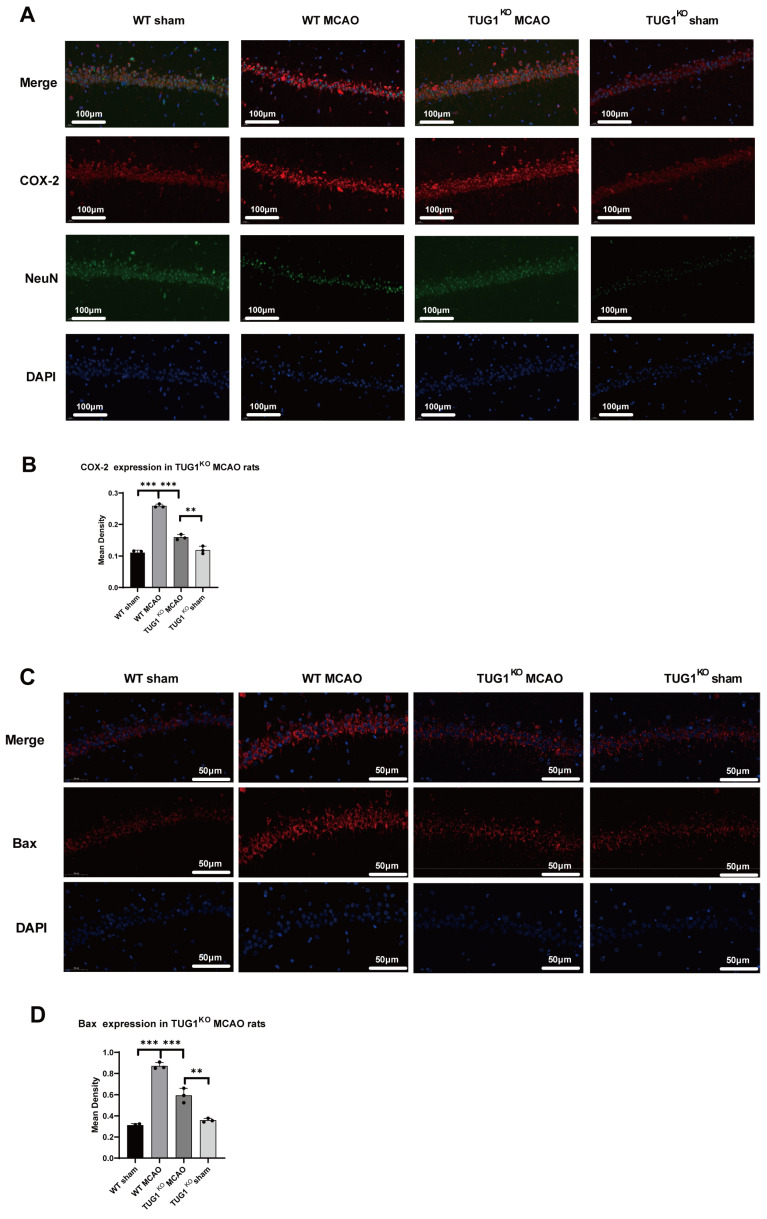
TUG1 improves COX-2 mRNA and Bax mRNA stability through HuR. (**A**) Immunofluorescence staining of COX-2 was conducted on neurons in the CA1 region of the hippocampus. (**B**) The fluorescent intensity of COX-2 was quantitatively analyzed. (** *p* < 0.01; *** *p* < 0.001) (**C**) Immunofluorescence staining of Bax was conducted on neurons in the CA1 region of the hippocampus. (**D**) The fluorescent intensity of Bax was quantitatively analyzed. Cell nuclei were stained with DAPI (blue). (** *p* < 0.01; *** *p* < 0.001).

**Table 1 biomedicines-12-02520-t001:** gRNA Sequence Table.

gRNA	Sequence
gRNA1	TCTCGTACGCAGAACTCGGGCGG
gRNA2	CTAACGTTTGCAATGCAATCAGG

**Table 2 biomedicines-12-02520-t002:** Table of TUG1 Primers.

Primer Name	Sequence
TUG1-F	ACGTGACCGGATCTTGTTTAGCC
TUG1-R	GCTTAGACTGCTTGAATCTTCGCCA
TUG1-F1	AGCAATTCTAAGGTGGCACTGTGGTAG
TUG1-R1	ACACTGGGTTAAATGAAGTCTTGCTGC
TUG1-He/Wt-F	TTAAAGTGACGGCTACTAAATCCTGATTG
TUG1-He/Wt-R	TCGTCGGATCGCAAAGGCATA

**Table 3 biomedicines-12-02520-t003:** PCR amplification products for TUG1 genotyping.

Region	F	R	Wild Type	Heterozygous (TUG1+/−)	Homozygous (TUG1−/−)
1	TUG1-F	TUG1-R	10263 bp	None	None
2	TUG1-F	TUG1-R TUG1-He/Wt-R	427 bp	777 bp and 427 bp	777 bp
3	TUG1-F TUG1-He/Wt-F	TUG1-R	424 bp	777 bp and 424 bp	777 bp
4	TUG1-F1	TUG1-R1	531 bp	531 bp	0 bp

## Data Availability

Data are contained within the article.

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
