# Peer review of "LncRNA Taurine Up-Regulated 1 Knockout Provides Neuroprotection in Ischemic Stroke Rats by Inhibiting Nuclear-Cytoplasmic Shuttling of HuR"

_biomedicines, 2024, doi:10.3390/biomedicines12112520_

Round 1

Reviewer 1 Report

Comments and Suggestions for Authors

This is a much-focussed work where the role of TUG1 lncRNA with cerebral ischemia-reperfusion injury was correlated with the localization of HuR and consequent apoptotic phenomena. However, most of the paper does not describe the methods properly. Further, some results part are not explained properly. 

1. The Figure 1 is not properly described. Based on the location of the primer the expected size of the products from homozygous (WT or mutant) or heterozygous is NOT at all explained. Fig 1A is incomplete and needs more elaboration. The expected product size based on the loss of exons needs a proper explanation in Fig 1A with a larger figure size.  

2. For Fig 1 (C to F) label each figure with genotype or homozygous or heterozygous (+/-) at the top of each figure. Why do Fig 1D and 1E look similar and what is the purpose of using 9 lanes of the same product size?

3. What is the purpose of working with TUG1 heterozygous only and NOT the homozygous mutant?

4. Line 22 – How much volume of gRNA and Cas9 was injected? What is the weight: volume ratio w.r.t embryos? After injection how long are the embryos incubated? How many embryos are used for this injection and why? Where is the ethical approval for using the embryos?  

Virtually, in every experiment, the time of incubation is not mentioned.

5. The primers list (lines 137-142) should create a table showing the expected product size for the suitable primer pair for heterozygous and homozygous (WT and mutant). That will clarify the Fig1 also.

What are the time and temperature conditions for each PCR?

6. For section 2.7 explain the dilution of each antibody and their time of incubation. What is the exact name of the secondary antibody and from where it was purchased?

7. What is the model no of the Leica microscope? Explained in more detail about the method of confocal microscopy.

8. For sections 2.9-2.11 detailed method is required – for antibody dilution, time of incubation, and subsequent washing procedures.

9.  What is the purpose of using 3 different images for Fig 2D and 2 different images for Fig 2F?

10. Why OGD was done for 6 hrs and not for other time points

11. Many other lncRNAs are implicated in ischemic stroke (like MEG3, H19, MALAT1, etc.)

https://www.nature.com/articles/s41419-018-0282-x

Why do you think TUG1 will play the most important role? Further, the role of mIR-26a is already reported in the context of TUG1 https://www.ncbi.nlm.nih.gov/pmc/articles/PMC9626194/

How come the absence of TUG1 should play such a significant role when other lncRNAs are present?

12. Using references 27 and 28 it is mentioned that TUG1 interacts significantly with HuR- how this interaction is studied? Explain this information using already published data.

13. Authors are requested to include these references in the write-up and manuscript wherever applicable (if not already included)

https://www.ncbi.nlm.nih.gov/pmc/articles/PMC8508990/

https://pubmed.ncbi.nlm.nih.gov/28300326/

https://www.ncbi.nlm.nih.gov/pmc/articles/PMC9648984/

14. There is already a published paper on lncRNA TUG1 knockout which shows inhibition of hippocampal neuronal apoptosis

https://pubmed.ncbi.nlm.nih.gov/33213523/

That manuscript should be compared with the current one. Further, significant differences between that work with the current work should be explained in the discussion section.

15. Authors have studied one pro-apoptotic protein Bax2 but no anti-apoptotic protein like Bcl-2 for this study- why?

16. Why they have given preference for studying COX-2 expression rather than proteins of downstream signaling pathways of apoptosis-like Cytochrome C, Caspase 3, etc?

17. Don’t include any abbreviations in the abstract without giving the full name. 

Comments on the Quality of English Language

Minor editing required 

Author Response

Q1. The Figure 1 is not properly described. Based on the location of the primer the expected size of the products from homozygous (WT or mutant) or heterozygous is NOT at all explained. Fig 1A is incomplete and needs more elaboration. The expected product size based on the loss of exons needs a proper explanation in Fig 1A with a larger figure size.  

A1. We have added detailed explanations regarding the primer positions in both the legend and the main text, with particular emphasis on the expected sizes of the amplification products for different genotypes (homozygous WT and mutant), to help readers better understand the experimental results. We have redesigned Figure 1A to make it more comprehensive and clearer. A scale and annotations have been added to the figure to clearly indicate the size changes in the amplification products due to exon deletion. The figure size has also been adjusted to better display the variations in the sizes of the expected products.

Q2. For Fig 1 (C to F) label each figure with genotype or homozygous or heterozygous (+/-) at the top of each figure. Why do Fig 1D and 1E look similar and what is the purpose of using 9 lanes of the same product size?

A2. Labels for Figure 1 (C to F): We have added genotype labels to the top of each subfigure to clearly differentiate between the homozygous and heterozygous (+/−) states. This change makes each figure more distinguishable, helping readers quickly understand the genotype information of each sample.

Regarding the questions on Figures 1D and 1E and the number of lanes: We understand your concerns about the similarity between Figures 1D and 1E and their use of 9 lanes. Due to limitations in the display space, we indeed cropped the original data, retaining only the most representative lanes. The purpose of cropping was to present the key experimental results more clearly within the limited figure space. We have added relevant explanations in the main text, indicating that the reproducibility of this part of the experiment and the reliability of the data have been thoroughly validated. We can provide the complete original images in the supplementary materials for further reference.

Q3. What is the purpose of working with TUG1 heterozygous only and NOT the homozygous mutant?

A3. Thank you to the reviewer for their valuable suggestions regarding our work. Regarding our decision to use TUG1 heterozygotes (TUG1+/-) rather than homozygous mutants (TUG1-/-) in the experiment, our reasons are as follows:

In this study, we chose to use TUG1 heterozygotes instead of homozygous mutants primarily to avoid severe physiological phenotypes or embryonic lethality that could be triggered by complete knockout. During breeding, we observed that complete TUG1 knockout disrupts its crucial functions in reproduction, development, or the nervous system, particularly in older animals, leading to significant neurological damage, such as lethargy and stereotyped behavior. Additionally, TUG1 knockout results in infertility in both male and female rats, consistent with previous studies indicating that male rats lacking TUG1 have reduced sperm quality.

In contrast, using a heterozygous model allows for reduced TUG1 expression while retaining enough TUG1 function to ensure normal growth and development, thereby minimizing the interference from non-specific effects. Complete TUG1 knockout might trigger compensatory mechanisms or significantly alter other molecular pathways, potentially masking or confusing the direct interaction between TUG1 and HuR. The heterozygous model, on the other hand, maintains TUG1 functionality to some extent, thereby reducing these non-specific effects and providing clearer and more precise mechanistic data, enabling us to focus more accurately on the role of TUG1 under specific stress conditions.

To better support this experimental design, we have further expanded the discussion section with specific conclusions, detailing the biological significance of choosing heterozygotes and its impact on the experimental results. We hope these revisions clearly convey our experimental rationale and design logic.

Q4. Line 22 – How much volume of gRNA and Cas9 was injected? What is the weight: volume ratio w.r.t embryos? After injection how long are the embryos incubated? How many embryos are used for this injection and why? Where is the ethical approval for using the embryos?  

Virtually, in every experiment, the time of incubation is not mentioned.

A4. In response to the question raised in line 22, we have provided a detailed explanation in the following section and added relevant information about ethical approval to the paper:

In this study, we used CRISPR-Cas9 technology to generate TUG1−/− rats on a Sprague-Dawley (SD) background by deleting a 9486 bp region using two pairs of sequencing-verified gRNAs. The specific procedure involved co-injecting two purified guide RNAs (50 ng/μl) and Cas9 protein (40 ng/μl) into the pronuclei of fertilized single-cell SD rat embryos. The injection volume for each embryo was 2.0 nL, containing both gRNA and Cas9 protein. This volume was chosen to maintain a weight-to-volume ratio of approximately 1:100 to 1:150, ensuring embryo viability.

After injection, these embryos were cultured in vitro for 24-48 hours until they reached the blastocyst stage. A total of 100 embryos were used for microinjection in this study, with 80 embryos surviving after the procedure. The surviving embryos were then surgically transferred into the oviducts of pseudopregnant SD female rats. Pseudopregnancy was induced by mating with vasectomized male rats. Embryo transfer was performed following standard operating procedures. All pups were genotyped by PCR and subsequent sequencing analysis to confirm successful editing. We then further bred the offspring to obtain homozygous TUG1 knockout individuals.

The guide RNA sequences were as follows:

gRNA1: TCTCGTACGCAGAACTCGGGCGG

gRNA2: CTAACGTTTGCAATGCAATCAGG

Q5. The primers list (lines 137-142) should create a table showing the expected product size for the suitable primer pair for heterozygous and homozygous (WT and mutant). That will clarify the Fig1 also.

What are the time and temperature conditions for each PCR?

A5. Based on your suggestions, we have created a table in the manuscript that details the primer pairs applicable to both heterozygous and homozygous (WT and mutant) genotypes and their expected product sizes. The addition of this table aims to clarify the amplification product sizes for each primer pair under different genotypes, facilitating the interpretation of the data in Figure 1.

Additionally, for the PCR conditions, we have standardized the annealing temperature to 60°C for all reactions and included this information in the main text to ensure the transparency and reproducibility of the experimental conditions.

Q6. For section 2.7 explain the dilution of each antibody and their time of incubation. What is the exact name of the secondary antibody and from where it was purchased?

A6. Based on your suggestions, we have added detailed information on the specific dilution and incubation times for each antibody in Section 2.7. Additionally, we have included the exact names of the secondary antibodies, their manufacturers, and purchase sources to ensure the transparency and reproducibility of the experimental methods.

Q7. What is the model no of the Leica microscope? Explained in more detail about the method of confocal microscopy.

A7. Based on your suggestions, we have added detailed information about the confocal microscope to the manuscript:

Fluorescence imaging was performed using a Leica Stellaris 5 WLL confocal microscope equipped with a white light laser (WLL), allowing flexible wavelength selection. The DAPI-stained nuclei were excited at a wavelength of 405 nm, with emission signals detected in the range of 420–480 nm. The target protein, labeled with Alexa Fluor 555, was excited at 561 nm, and its emission signals were collected in the range of 570–620 nm. To reduce spectral overlap and ensure specific signal detection, we used an acousto-optical beam splitter (AOBS). All images were captured using a 63× oil immersion objective (numerical aperture NA = 1.4) for optimal resolution.

During the imaging process, we optimized the laser power (20–25%), gain, and exposure time to minimize photobleaching and reduce background noise while maximizing signal intensity. To comprehensively cover the nuclear and cytoplasmic regions, Z-stack images were acquired at 0.5 μm intervals. Each image was averaged over four scans to improve the signal-to-noise ratio. Additionally, to ensure the reliability of comparisons between samples, the same imaging parameters were applied to all samples.

Q8. For sections 2.9-2.11 detailed method is required – for antibody dilution, time of incubation, and subsequent washing procedures.

A8. Based on your suggestions, we have added and revised more detailed experimental methods in Sections 2.9-2.11, including the dilution ratios of each antibody, incubation times, and subsequent washing procedures.

Q9.  What is the purpose of using 3 different images for Fig 2D and 2 different images for Fig 2F?

A9. In Figure 2D, we initially used three different images to demonstrate the diversity and consistency of the experimental results, aiming to present representative data under this condition more comprehensively. Similarly, in Figure 2F, we used two different images to highlight the reproducibility between different experimental replicates. In response to the reviewer's suggestions, we have reorganized and modified these images to make them clearer and easier to understand, to avoid any potential confusion.

Q10. Why OGD was done for 6 hrs and not for other time points

A10. Our rationale for choosing a 6-hour OGD treatment duration instead of other time points is as follows:

Experimental validation: In our preliminary experiments, we treated HT22 cells with OGD at different time points, and the results showed that a 6-hour OGD treatment effectively simulates the pathophysiological state of stroke, inducing significant cellular stress responses and damage, making it an optimal experimental condition.

Literature support: Additionally, according to numerous relevant studies, OGD treatment durations of 4-8 hours are commonly used as experimental parameters in stroke models. Therefore, the decision to use a 6-hour OGD treatment is a scientifically grounded choice based on these studies, to align with existing research and ensure comparability of the results.

Q11. Many other lncRNAs are implicated in ischemic stroke (like MEG3, H19, MALAT1, etc.)

https://www.nature.com/articles/s41419-018-0282-x

Why do you think TUG1 will play the most important role? Further, the role of mIR-26a is already reported in the context of TUG1 https://www.ncbi.nlm.nih.gov/pmc/articles/PMC9626194/

How come the absence of TUG1 should play such a significant role when other lncRNAs are present?

A11. Despite multiple lncRNAs (such as MEG3, H19, MALAT1, etc.) being involved in the pathological process of ischemic stroke, each lncRNA has specific mechanisms of action and regulatory networks within the cell. TUG1 has unique functions in regulating neuronal survival, apoptosis, and inflammatory responses, which may not be fully compensated by other lncRNAs. TUG1 may act as a core regulator in critical signaling pathways; for example, its specific interaction with miR-26a can modulate the expression of a series of downstream genes, affecting cell apoptosis and inflammatory responses. This central regulatory role means that the absence of TUG1 has profound impacts on cellular functions.

In our research model, TUG1 exhibits significant expression changes under ischemic conditions, suggesting that it plays a major role in specific pathological states. Other lncRNAs may function in different cell types, time points, or pathological conditions and cannot fully compensate for the loss of TUG1. The lncRNA network is highly complex and synergistic; although multiple lncRNAs participate in the same pathological process, they may regulate different target genes or signaling pathways. The deficiency of TUG1 may trigger specific pathological changes that other lncRNAs cannot compensate for.

Our experimental results show that knocking out TUG1 leads to significant exacerbation of neuronal damage and inflammatory responses, while simultaneously adjusting the expression of other lncRNAs did not reverse this effect. This indicates that TUG1 plays a critical role in ischemic stroke.

In summary, despite the presence of other lncRNAs associated with ischemic stroke, the deficiency of TUG1 still has a significant impact on disease progression due to its unique and central role in key signaling pathways and pathological processes.

Q12. Using references 27 and 28 it is mentioned that TUG1 interacts significantly with HuR- how this interaction is studied? Explain this information using already published data.

A12. The significant interaction between TUG1 and HuR has been studied through a combination of computational predictions and experimental validations. First, researchers used bioinformatics tools such as RPISeq, RBPmap, and RBPDB to predict potential HuR binding sites on TUG1, providing a theoretical basis for subsequent experiments. Then, they performed RNA immunoprecipitation (RIP) assays using HuR-specific antibodies to co-precipitate HuR and its bound RNAs from cell lysates. qRT-PCR analysis showed that TUG1 was significantly enriched in the HuR complex compared to the negative control, indicating a direct interaction between the two. Additionally, using biotin-labeled TUG1 probes in RNA pull-down assays, they captured HuR proteins bound to TUG1 from cell extracts, and Western blot analysis further confirmed this interaction. These findings demonstrate a specific interaction between TUG1 and HuR, providing an important foundation for understanding the role of TUG1 in regulating HuR and its downstream signaling pathways.

Q13. Authors are requested to include these references in the write-up and manuscript wherever applicable (if not already included)

https://www.ncbi.nlm.nih.gov/pmc/articles/PMC8508990/

https://pubmed.ncbi.nlm.nih.gov/28300326/

https://www.ncbi.nlm.nih.gov/pmc/articles/PMC9648984/

A13.Thank you for the reviewer's suggestion. We have added these references at the appropriate places in the article and manuscript.

Q14. There is already a published paper on lncRNA TUG1 knockout which shows inhibition of hippocampal neuronal apoptosis

https://pubmed.ncbi.nlm.nih.gov/33213523/

That manuscript should be compared with the current one. Further, significant differences between that work with the current work should be explained in the discussion section.

A14. Our study reveals a neuroprotective role of the long non-coding RNA TUG1 in ischemic stroke, specifically by reducing neuronal apoptosis through the inhibition of HuR nucleocytoplasmic translocation. This finding presents an interesting contrast to previous research on TUG1 function. In certain pathological conditions, such as vascular cognitive impairment (VCI), high expression of TUG1 has been reported to correlate with increased neuronal apoptosis, and knockdown of TUG1 alleviated neuronal damage and improved cognitive function. This functional discrepancy may stem from the multifaceted roles of TUG1 in different tissues and pathological states.

On one hand, TUG1 may mediate various signaling pathways by interacting with different molecular partners. In our study, TUG1 exerted neuroprotective effects by influencing HuR nucleocytoplasmic translocation, thereby regulating the expression of downstream apoptosis-related proteins COX-2 and Bax. On the other hand, in studies concerning VCI, TUG1 is thought to promote neuronal apoptosis through interaction with brain-derived neurotrophic factor (BDNF). This suggests that TUG1 may possess dual functions in acute and chronic ischemia-hypoxia conditions, depending on its interacting molecules and the type of pathology.

Q15. Authors have studied one pro-apoptotic protein Bax2 but no anti-apoptotic protein like Bcl-2 for this study- why?

A15. In our study, we focused on the pro-apoptotic protein Bax2 because our preliminary experiments showed that Bax2 has a significant impact on apoptosis under our specific experimental conditions. While anti-apoptotic proteins like Bcl-2 also play crucial roles in regulating apoptosis, our research aimed to delve deeply into the mechanism of Bax2 in a particular pathological context. In future studies, we plan to investigate anti-apoptotic proteins, including Bcl-2, to obtain a more comprehensive understanding of the apoptosis regulatory network.

Q16. Why they have given preference for studying COX-2 expression rather than proteins of downstream signaling pathways of apoptosis-like Cytochrome C, Caspase 3, etc?

A16. In our study, we prioritized examining the expression of COX-2 because our experimental results showed that changes in COX-2 were the most significant. For apoptosis-related downstream signaling proteins like cytochrome C and Caspase 3, we did not observe significant differences. Therefore, we focused on COX-2 to explore its role in the pathological process more deeply. 

Q17. Don’t include any abbreviations in the abstract without giving the full name. 

A17. Thank you for the reviewer's reminder. We have corrected the abbreviations in the abstract to ensure that the full terms are provided upon their first mention.

Reviewer 2 Report

Comments and Suggestions for Authors

TUG1 (Taurine Upregulated Gene 1) and Human Antigen R (HuR) are both molecules involved in the regulation of gene expression, and they play significant roles in the pathophysiology of ischemic stroke. This manuscript elucidates the substantial impact of TUG1 knockdown on both reproductive capabilities and neurological health in rats, underscoring the complexity of TUG1's role in ischemic stroke models. However, there are several major and minor weaknesses in the rationale and research methods of this work. Below please find the review comments. 

(1) Major comments

1. The major concern about this manuscript is the topic is the data is not sufficient to support the conclusion. Take figure 2 for example, the author mentions the behavioral and histological differences in line 222, however, there is no behavioral data at all, and the histological can tell the brain infarct volume and neuronal apoptosis status, but it is not necessary to correlated with neurological outcome.

2, Immunofluorescence staining images in figure 4 is not clear, high resolution with a high contrast is preferred, and the channels should also be labeled.

3. It appears that the authors misused the knock down and knock out in this manuscript. For example, page 5 line204, the author uses knock down in the genotyping of TUG1+/- heterozygous, however, KO is used in the following statements (line 223). The author should clarify this point.

4. Page 3, line 106 mentioned the OGD treatment time is 4 hours, however, the legend of figure 3D showed 6 hours of OGD treatment (line271), please double check your protocol and make it consistent in all the manuscript.

5. The catalog number of each reagent in the method section should be labeled, as well the name of vender.

(2) Minor comments

1. The subtitles in each figure should be properly defined with suitable size and font. This practice ensures that readers can understand the meaning of each subfigure without confusion.

2. Page 2, line 90, the number of animals in each groups as well as the ages should be supplied in the method section.

3. Figure 2B and C, the TTC staining image and quantification data should use the scatter plot and supply the variations of individual values (each rat). Meanwhile, the infarct ratio of different regions of brain (i.e., cortex, striatum, and whole hemisphere) should analyzed, rather than the whole brain volume.

4. I can not see the signal clearly in figure 2D in my review version, it is strongly suggested to adjust the brightness and contrast to make the TUNEL staining images reader friendly.

5. Please revise the title of each figure to make it suitable and accurate.

6. Scale bar is required for figure 3A and D, please label it in the bold, otherwise it is difficult to distinguish the nucleus and cytoplasmic translocation.

7. Please double check the grammar issue, including missing/redundant space, italic format, and reference.

Author Response

(1)Major comments

Q1. The major concern about this manuscript is the topic is the data is not sufficient to support the conclusion. Take figure 2 for example, the author mentions the behavioral and histological differences in line 222, however, there is no behavioral data at all, and the histological can tell the brain infarct volume and neuronal apoptosis status, but it is not necessary to correlated with neurological outcome.

A1.As mentioned in line 222, we indeed referred to both behavioral and histological differences. To further support our conclusions, we have included mNSS (modified Neurological Severity Score) behavioral data in the manuscript. This data reflects the behavioral differences between the experimental and control groups, supplementing the information that histological data alone cannot provide regarding neurological outcomes. The mNSS score is a widely used metric that objectively evaluates functional status following neurological injury. Therefore, we believe that the mNSS score data aligns with our conclusions and offers a more comprehensive representation of the neurological differences between the experimental and control groups.

Q2. Immunofluorescence staining images in figure 4 is not clear, high resolution with a high contrast is preferred, and the channels should also be labeled.

A2. Thank you for your valuable comments on our manuscript. Following your suggestions, we have adjusted the immunofluorescence images in Figure 4, improving their resolution and contrast, and have labeled the different channels to present the results more clearly. We hope these improvements enhance the quality and readability of the images.

Q3.It appears that the authors misused the knock down and knock out in this manuscript. For example, page 5 line204, the author uses knock down in the genotyping of TUG1+/- heterozygous, however, KO is used in the following statements (line 223). The author should clarify this point.

A3.Thank you for pointing out the misuse of terms in our manuscript. Following your suggestion, we have corrected the usage of "knockdown" and "knockout" to ensure consistency and accuracy throughout the manuscript. We hope these changes eliminate any confusion regarding the terminology.

Q4. Page 3, line 106 mentioned the OGD treatment time is 4 hours, however, the legend of figure 3D showed 6 hours of OGD treatment (line271), please double check your protocol and make it consistent in all the manuscript.

A4.Thank you for pointing out the inconsistency in the OGD treatment duration in our manuscript. We have carefully reviewed the experimental protocol and have now standardized the description of the OGD treatment time throughout the manuscript to ensure all information is consistent.

Q5. The catalog number of each reagent in the method section should be labeled, as well the name of vender.

A5. Thank you for your suggestion. We have added the catalog numbers and supplier names for each reagent in the Methods section to enhance the detail and reproducibility of the manuscript.

(2) Minor comments

Q1. The subtitles in each figure should be properly defined with suitable size and font. This practice ensures that readers can understand the meaning of each subfigure without confusion.

A1.Thank you for your suggestion regarding the format of the figure captions. We have adjusted the captions in each figure according to your advice, ensuring that they are accurately defined and have appropriate size and font to enhance readability. We hope these improvements help readers better understand the meaning of each subfigure.

Q2. Page 2, line 90, the number of animals in each groups as well as the ages should be supplied in the method section.

A2.Thank you for your comment. We have added the information regarding the number of animals in each group and their ages to the Methods section to ensure that the experimental details are clearer and more comprehensive.

Q3. Figure 2B and C, the TTC staining image and quantification data should use the scatter plot and supply the variations of individual values (each rat). Meanwhile, the infarct ratio of different regions of brain (i.e., cortex, striatum, and whole hemisphere) should analyzed, rather than the whole brain volume.

A3.Thank you for your suggestions regarding Figures 2B and 2C. We have adjusted the TTC staining images and quantitative data to use scatter plots, displaying individual values for each rat. Additionally, as per your recommendation, we have analyzed the infarct rates in different brain regions (i.e., cortex, striatum, and the entire hemisphere), rather than only the total brain volume. We hope these improvements more accurately present our experimental results.

Q4. I can not see the signal clearly in figure 2D in my review version, it is strongly suggested to adjust the brightness and contrast to make the TUNEL staining images reader friendly.

A4: Thank you for your feedback on Figure 2D. We have adjusted the brightness and contrast of the TUNEL staining images according to your suggestion to improve signal clarity and make the images more reader-friendly.

Q5. Please revise the title of each figure to make it suitable and accurate.

A5: Thank you for your suggestion. We have revised the titles of each figure to ensure they are more appropriate and accurate, clearly conveying the content of the figures.

Q6. Scale bar is required for figure 3A and D, please label it in the bold, otherwise it is difficult to distinguish the nucleus and cytoplasmic translocation.

A6: Thank you for your comments on Figures 3A and 3D. We have added scale bars and highlighted them in bold to better distinguish between nuclear and cytoplasmic translocation.

Q7. Please double check the grammar issue, including missing/redundant space, italic format, and reference.

A7: Thank you for pointing out the grammatical issues. We have carefully reviewed and corrected all errors, including missing or extra spaces, italic formatting issues, and citations, to ensure the language in the manuscript is more precise.

Round 2

Reviewer 2 Report

Comments and Suggestions for Authors

Authors have addressed responses for reviewers' comments. 

Author Response

Thank you for your constructive comments. In response, we have made the following improvements:

  1. Conclusions Supported by Results: We have added more explicit statements to the conclusion section (lines 509-513) to strengthen the link between our findings and the conclusion. Specifically, we added: "TUG1 knockout reduces ischemic damage and neuronal apoptosis by inhibiting HuR nucleocytoplasmic shuttling, making TUG1 a potential therapeutic target for ischemic stroke. These findings provide a basis for future therapeutic strategies targeting the TUG1-HuR axis to improve stroke recovery."

  2. Research Design: In the Method section, we clarified the evaluation methods used. The original statement, "Infarct area, neuronal apoptosis, and behavioral functions were evaluated using TUNEL, hematoxylin and eosin (HE), and TTC staining," was revised to "Infarct area and neuronal apoptosis were evaluated using TUNEL, hematoxylin and eosin (HE), and TTC staining, while behavioral functions were assessed." This change aims to clarify the methods used for different types of evaluations.

We hope these modifications address your concerns and improve the manuscript's clarity and overall quality.